# MULTI-HEAD ATTENTION:
# COLLABORATE INSTEAD OF CONCATENATE

## ABSTRACT

Attention layers are widely used in natural language processing (NLP) and are beginning to influence computer vision architectures. Training very large transformer models allowed significan improvement in both fields, but once trained, these networks show symptoms of over-parameterization. For instance, it is known that many attention heads can be pruned without impacting accuracy. This work aims to enhance current understanding on how multiple heads interact. Motivated by the observation that trained attention heads share common key/query projections, we propose a collaborative multi-head attention layer that enables heads to learn shared projections. Our scheme decreases the number of parameters in an attention layer and can be used as a drop-in replacement in any transformer architecture. For instance, by allowing heads to collaborate on a neural machine translation task, we can reduce the key dimension by $4\times$ without any loss in performance. We also show that it is possible to re-parametrize a pre-trained multi-head attention layer into our collaborative attention layer. Even without retraining, collaborative multi-head attention manages to reduce the size of the key and query projections by half without sacrificing accuracy. Our code is public.[1]

## 1 INTRODUCTION

Since the invention of attention (Bahdanau et al., 2014) and its popularization in the transformer architecture (Vaswani et al., 2017), multi-head attention (MHA) has become the de facto architecture for natural language understanding tasks (Devlin et al., 2019) and neural machine translation. Attention mechanisms have also gained traction in computer vision following the work of Ramachandran et al. (2019) and Bello et al. (2019). Nevertheless, despite their wide adoption, we currently lack solid theoretical understanding of how transformers operate. In fact, many of their modules and hyperparameters are derived from empirical evidences that are possibly circumstantial.

The uncertainty is amplified in multi-head attention, where both the roles and interactions between heads are still poorly understood. Empirically, it is well known that using multiple heads can improve model accuracy. However, not all heads are equally informative, and it has been shown that certain heads can be pruned without impacting model performance. For instance, Voita et al. (2019) present a method to quantify head utility and prune redundant members. Michel et al. (2019) go further to question the utility of multiple heads by testing the effect of heavy pruning in several settings. On the other hand, Cordonnier et al. (2020) prove that multiple heads are needed for self-attention to perform convolution, specifically requiring one head per pixel in the filter's receptive field. Beyond the number of heads, finding the adequate head dimension is also an open question. Bhojanapalli et al. (2020) finds that the division of the key/query projection between heads gives rise to a low-rank bottleneck for each attention head expressivity that can be fixed by increasing the head sizes. In contrast, our approach increases heads expressivity by leveraging the low-rankness accross heads to share common query/key dimensions.

This work aims to better detect and quantify head redundancy by asking whether independent heads learn overlapping or distinct concepts. This relates to the work on CNN compression that factorizes common filters in a trained convolutional network (Kim et al., 2016) using Tucker decomposition. In attention models, we discover that some key/query projected dimensions are redundant, as trained

---

[1] https://github.com/...

concatenated heads tend to compute their attention patterns on common features. Our finding implies that MHA can be re-parametrized with better weight sharing for these common projections and a lower number of parameters. This differs from concurrent work (Shazeer et al., 2020) that orchestrate collaboration between heads on top of the dot product attention scores.

*Contribution 1: Introducing the collaborative multi-head attention layer.* Section 3 describes a collaborative attention layer that allows heads to learn shared key and query features. The proposed re-parametrization significantly decreases the number of parameters of the attention layer without sacrificing performance. Our Neural Machine Translation experiments in Section 4 show that the number of FLOPS and parameters to compute the attention scores can be divided by 4 without affecting the BLEU score on the WMT14 English-to-German task.

*Contribution 2: Re-parametrizing pre-trained models into a collaborative form renders them more efficient.* Pre-training large language models has been central to the latest NLP developments. But pre-training transformers from scratch remains daunting for its computational cost even when using more efficient training tasks such as (Clark et al., 2020). Interestingly, our changes to the MHA layers can be applied post-hoc on pre-trained transformers, as a drop-in replacement of classic attention layers. To achieve this, we compute the weights of the re-parametrized layer using canonical tensor decomposition of the query and key matrices in the original layer. Our experiments in Section 4 show that the key/query dimensions can be divided by 3 without any degradation in performance.

As a side contribution, we identify a discrepancy between the theory and some implementations of attention layers and show that by correctly modeling the biases of key and query layers, we can clearly differentiate between context and content-based attention.

## 2 MULTI-HEAD ATTENTION

We first review standard multi-head attention introduced by Vaswani et al. (2017).

### 2.1 ATTENTION

Let $\boldsymbol{X} \in \mathbb{R}^{T \times D_{in}}$ and $\boldsymbol{Y} \in \mathbb{R}^{T' \times D_{in}}$ be two input matrices consisting of respectively $T$ and $T'$ tokens of $D_{in}$ dimensions each. An attention layer maps each of the $T$ query token from $D_{in}$ to $D_{out}$ dimensions as follows:

$$\text{Attention}(\boldsymbol{Q}, \boldsymbol{K}, \boldsymbol{V}) = \text{softmax}\left(\frac{\boldsymbol{Q}\boldsymbol{K}^{\top}}{\sqrt{d_k}}\right)\boldsymbol{V}, \text{ with } \boldsymbol{Q} = \boldsymbol{X}\boldsymbol{W}_Q, \boldsymbol{K} = \boldsymbol{Y}\boldsymbol{W}_K, \boldsymbol{V} = \boldsymbol{Y}\boldsymbol{W}_V \quad (1)$$

The layer is parametrized by a query matrix $\boldsymbol{W}_Q \in \mathbb{R}^{D_{in} \times D_k}$, a key matrix $\boldsymbol{W}_K \in \mathbb{R}^{D_{in} \times D_k}$ and a value matrix $\boldsymbol{W}_V \in \mathbb{R}^{D_{in} \times D_{out}}$. Using attention on the same sequence (i.e. $\boldsymbol{X} = \boldsymbol{Y}$) is known as self-attention and is the basic building block of the transformer architecture.

### 2.2 CONTENT VS. CONTEXT

Some re-implementations of the original transformer architecture[2] use biases in the linear layers. This differs from the attention operator defined in eq. (1) where the biases $\boldsymbol{b}_Q$ and $\boldsymbol{b}_K \in \mathbb{R}^{D_k}$ are ommited. Key and query projections are computed as $\boldsymbol{K} = \boldsymbol{X}\boldsymbol{W}_K + \mathbf{1}_{T \times 1}\boldsymbol{b}_K$ and $\boldsymbol{Q} = \boldsymbol{Y}\boldsymbol{W}_Q + \mathbf{1}_{T \times 1}\boldsymbol{b}_Q$, respectively, where $\mathbf{1}_{a \times b}$ is an all one matrix of dimension $a \times b$. The exact computation of the (unscaled) attention scores can be decomposed as follows:

$$\boldsymbol{Q}\boldsymbol{K}^{\top} = (\boldsymbol{X}\boldsymbol{W}_Q + \mathbf{1}_{T \times 1}\boldsymbol{b}_Q^{\top})(\boldsymbol{Y}\boldsymbol{W}_K + \mathbf{1}_{T \times 1}\boldsymbol{b}_K^{\top})^{\top} \quad (2)$$

$$= \underbrace{\boldsymbol{X}\boldsymbol{W}_Q\boldsymbol{W}_K^{\top}\boldsymbol{Y}^{\top}}_{\text{context}} + \underbrace{\mathbf{1}_{T \times 1}\boldsymbol{b}_Q^{\top}\boldsymbol{W}_K^{\top}\boldsymbol{Y}^{\top}}_{\text{content}} + \boldsymbol{X}\boldsymbol{W}_Q\boldsymbol{b}_K\mathbf{1}_{1 \times T} + \mathbf{1}_{T \times T}\boldsymbol{b}_Q^{\top}\boldsymbol{b}_K \quad (3)$$

As the last two terms of eq. (3) have a constant contribution over all entries of the same row, they do not contribute to the computed attention probabilities (softmax is shift invariant and $\text{softmax}(\boldsymbol{x} + c) = \text{softmax}(\boldsymbol{x}), \forall c$). On the other hand, the first two terms have a clear meaning: $\boldsymbol{X}\boldsymbol{W}_Q\boldsymbol{W}_K^{\top}\boldsymbol{Y}^{\top}$

---

[2]For instance: the BERT orignal implementation, its HuggingFace re-implementation and FairSeq encoder-decoder transformer.

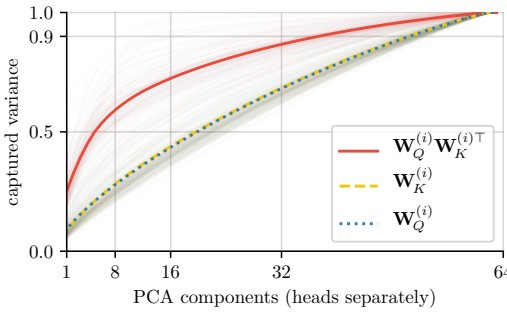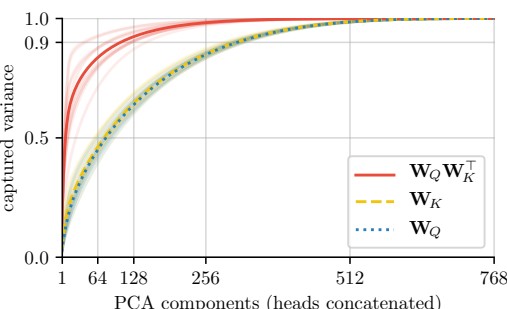

Figure 1: Cumulative captured variance of the key query matrices per head separately (*left*) and per layer with concatenated heads (*right*). Matrices are taken from a pre-trained BERT-base model with $N_h = 12$ heads of dimension $d_k = 64$. Bold lines show the means. Even though, by themselves, heads are not low rank (*left*), the product of their concatenation $\boldsymbol{W}_Q \boldsymbol{W}_K^\top$ is low rank (*right, in red*). Hence, the heads are sharing common projections in their column-space.

considers the relation between keys and query pairs, whereas $\mathbf{1}_{T \times 1} \boldsymbol{b}_Q^\top \boldsymbol{W}_K^\top \boldsymbol{Y}^\top$ computes attention solely based on key content.

The above findings suggest that the bias $\boldsymbol{b}_K$ of the key layer can be always be disabled without any consequence. Moreover, the query biases $\boldsymbol{b}_Q$ play an additional role: they allow for attention scores that are content-based, rather than solely depending on key-query interactions. This could provide an explanation for the recent success of the Dense-SYNTHESIZER (Tay et al., 2020), a method that ignores context and computes attention scores solely as a function of individual tokens. That is, perhaps context is not always crucial for attention scores, and content can suffice.

## 2.3 MULTI-HEAD ATTENTION

Traditionally, the attention mechanism is replicated by concatenation to obtain multi-head attention defined for $N_h$ heads as:

$$\text{MultiHead}(\boldsymbol{X}, \boldsymbol{Y}) = \underset{i \in [N_h]}{\text{concat}} \left[ \boldsymbol{H}^{(i)} \right] \boldsymbol{W}^O \tag{4}$$

$$\boldsymbol{H}^{(i)} = \text{Attention}(\boldsymbol{X}\boldsymbol{W}_Q^{(i)}, \boldsymbol{Y}\boldsymbol{W}_K^{(i)}, \boldsymbol{Y}\boldsymbol{W}_V^{(i)}), \tag{5}$$

where distinct parameter matrices $\boldsymbol{W}_Q^{(i)}, \boldsymbol{W}_K^{(i)} \in \mathbb{R}^{D_{in} \times d_k}$ and $\boldsymbol{W}_V^{(i)} \in \mathbb{R}^{D_{in} \times d_{out}}$ are learned for each head $i \in [N_h]$ and the extra parameter matrix $\boldsymbol{W}^O \in \mathbb{R}^{N_h d_{out} \times D_{out}}$ projects the concatenation of the $N_h$ head outputs (each in $\mathbb{R}^{d_{out}}$) to the output space $\mathbb{R}^{D_{out}}$. In the multi-head setting, we call $d_k$ the dimension of each head and $D_k = N_h d_k$ the total dimension of the query/key space.

## 3 IMPROVING THE MULTI-HEAD MECHANISM

Head concatenation is a simple and remarkably practical setup that gives empirical improvements. However, we show that another path could have been taken instead of concatenation. As the multiple heads are inherently solving similar tasks, they can collaborate instead of being independent.

### 3.1 HOW MUCH DO HEADS HAVE IN COMMON?

We hypothesize that some heads might attend on similar features in the input space, for example computing high attention on the verb of a sentence or extracting some dimensions of the positional encoding. To verify this hypothesis, it does not suffice to look at the similarity between query (or key) matrices $\{\boldsymbol{W}_Q^{(i)}\}_{i \in [N_h]}$ of different heads. To illustrate this issue, consider the case where two heads are computing the same key/query representations up to a unitary matrix $\boldsymbol{R} \in \mathbb{R}^{d_k \times d_k}$ such that

$$\boldsymbol{W}_Q^{(2)} = \boldsymbol{W}_Q^{(1)} \boldsymbol{R} \quad \text{and} \quad \boldsymbol{W}_K^{(2)} = \boldsymbol{W}_K^{(1)} \boldsymbol{R}.$$

Even though the two heads are computing identical attention scores, i.e. $\boldsymbol{W}_Q^{(1)}\boldsymbol{R}\boldsymbol{R}^\top\boldsymbol{W}_K^{(1)\top} = \boldsymbol{W}_Q^{(1)}\boldsymbol{W}_K^{(1)\top}$, they can have orthogonal column-spaces and the concatenation $[\boldsymbol{W}_Q^{(1)}, \boldsymbol{W}_Q^{(2)}] \in \mathbb{R}^{D_{in}\times 2d_k}$ can be full rank.

To disregard artificial differences due to common rotations or scaling of the key/query spaces, we study the similarity of the product $\boldsymbol{W}_Q^{(i)}\boldsymbol{W}_K^{(i)\top} \in \mathbb{R}^{D_{in}\times D_{in}}$ across heads. Figure 1 shows the captured energy by the principal components of the key, query matrices and their product. It can be seen on the left that single head key/query matrices $\boldsymbol{W}_Q^{(i)}\boldsymbol{W}_K^{(i)\top}$ are not low rank on average. However, as seen on the right, even if parameter matrices taken separately are not low rank, their concatenation is indeed low rank. This means that heads, though acting independently, learn to focus on the same subspaces. The phenomenon is quite pronounced: one third of the dimensions suffices to capture almost all the energy of $\boldsymbol{W}_Q\boldsymbol{W}_K^\top$, which suggests that there is inefficiency in the way multi-head attention currently operate.

## 3.2 Collaborative Multi-Head Attention

Following the observation that heads' key/query projections learn redundant projections, we propose to learn key/query projections for all heads at once and to let each head use a re-weighting of these projections. Our collaborative head attention is defined as follows:

$$\text{CollabHead}(\boldsymbol{X}, \boldsymbol{Y}) = \underset{i\in[N_h]}{\text{concat}}\left[\boldsymbol{H}^{(i)}\right]\boldsymbol{W}_O \tag{6}$$

$$\boldsymbol{H}^{(i)} = \text{Attention}(\boldsymbol{X}\tilde{\boldsymbol{W}}_Q\,\text{diag}(\boldsymbol{m}_i), \boldsymbol{Y}\tilde{\boldsymbol{W}}_K, \boldsymbol{Y}\boldsymbol{W}_V^{(i)})\,. \tag{7}$$

The main difference with standard multi-head attention defined in eq. (5) is that we do not duplicate the key and query matrices for each head. Instead, each head learns a mixing vector $\boldsymbol{m}_i \in \mathbb{R}^{\tilde{D}_k}$ that defines a custom dot product over the $\tilde{D}_k$ projected dimensions of the shared matrices $\tilde{\boldsymbol{W}}_Q$ and $\tilde{\boldsymbol{W}}_K$ of dimension $D_{in} \times \tilde{D}_k$. This approach leads to:

 (i) adaptive head expressiveness, with heads being able to use more or fewer dimensions according to attention pattern complexity;

 (ii) parameter efficient representation, as learned projections are shared between heads, hence stored and learned only once.

It is instructive to observe how standard multi-head attention (where heads are simply concatenated) can be seen as a special case of our collaborative framework (with $\tilde{D}_k = N_h d_k$). The left of Figure 2 displays the standard attention computed between $\boldsymbol{x}_n$ and $\boldsymbol{y}_m$ input vectors with the mixing matrix

$$\boldsymbol{M} := \underset{i\in[N_h]}{\text{concat}}\left[\boldsymbol{m}_i\right] \in \mathbb{R}^{N_h\times\tilde{D}_k}\,, \tag{8}$$

laying out the mixing vectors $\boldsymbol{m}_i$ as rows. In the concatenated MHA, the mixing vector $\boldsymbol{m}_i$ for the $i$-th head is a vector with ones aligned with the $d_k$ dimensions allocated to the $i$-th head among the $D_k = N_h d_k$ total dimensions.

Some alternative collaborative schema can be seen on the right side of Figure 2. By learning the mixing vectors $\{\boldsymbol{m}_i\}_{i\in[N_h]}$ instead of fixing them to this "blocks-of-1" structure, we increase the expressive power of each head for a negligible increase in the number of parameters. The size $d_k$ of each head, arbitrarily set to 64 in most implementations, is now adaptive and the heads can attend to a smaller or bigger subspace if needed.

## 3.3 Head Collaboration as Tensor Decomposition

As we show next, there is a simple way to convert any standard attention layer to collaborative attention without retraining. To this end, we must extract the common dimensions between query/key matrices $\{\boldsymbol{W}_Q^{(i)}\boldsymbol{W}_K^{(i)\top} \in \mathbb{R}^{D_{in}\times D_{in}}\}_{i\in[N_h]}$ across the different heads. This can be solved using the Tucker tensor decomposition (Tucker, 1966) of the 3rd-order tensor

$$\boldsymbol{\mathsf{W}}_{QK} := \underset{i\in[N_h]}{\text{stack}}\left[\boldsymbol{W}_Q^{(i)}\boldsymbol{W}_K^{(i)\top}\right] \in \mathbb{R}^{N_h\times D_{in}\times D_{in}}\,. \tag{9}$$

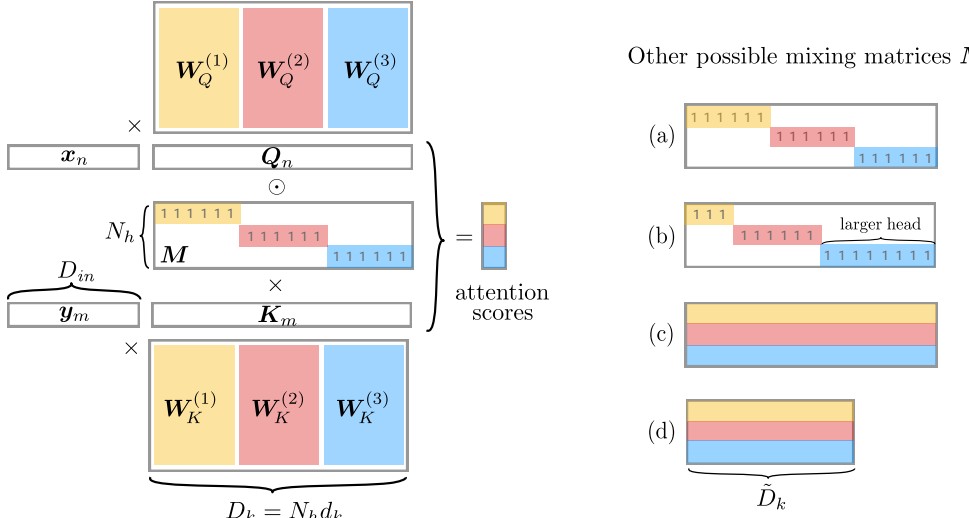

Figure 2: *Left*: computation of the attention scores between tokens $\boldsymbol{x}_n$ and $\boldsymbol{y}_m$ using a standard concatenated multi-head attention with $N_h = 3$ independent heads. The block structure of the mixing matrix $\boldsymbol{M}$ enforces that each head dot products non overlapping dimensions. *Right*: we propose to use more general mixing matrices $\boldsymbol{M}$ than (a) heads concatenation, such as (b) allowing heads to have different sizes; (c) sharing heads projections by learning the full matrix; (d) compressing the number of projections from $D_k$ to $\tilde{D}_k$ as heads can share redundant projections.

Following the notation[3] of Kolda & Bader (2009), the Tucker decomposition of a tensor $\mathbf{T} \in \mathbb{R}^{I \times J \times K}$ is written as

$$\mathbf{T} \approx \mathbf{G} \times_1 \boldsymbol{A} \times_2 \boldsymbol{B} \times_3 \boldsymbol{C} = \sum_{p=1}^{P} \sum_{q=1}^{Q} \sum_{r=1}^{R} g_{pqr} \, \boldsymbol{a}_p \circ \boldsymbol{b}_q \circ \boldsymbol{c}_r =: [\![\mathbf{G}; \boldsymbol{A}, \boldsymbol{B}, \boldsymbol{C}]\!], \qquad (10)$$

with $\boldsymbol{A} \in \mathbb{R}^{I \times P}$, $\boldsymbol{B} \in \mathbb{R}^{J \times Q}$, and $\boldsymbol{C} \in \mathbb{R}^{K \times R}$ being factor matrices, whereas $\mathbf{G} \in \mathbb{R}^{P \times Q \times R}$ is the core tensor. Intuitively, the core entry $g_{pqr} = \mathbf{G}_{p,q,r}$ quantifies the level of interaction between the components $\boldsymbol{a}_p, \boldsymbol{b}_q$, and $\boldsymbol{c}_r$.

In the case of attention, it suffices to consider the dot product of the aligned key/query components of the $\boldsymbol{Q}$ and $\boldsymbol{K}$ matrices, which means that the core tensor is super-diagonal (i.e. $g_{pqr} \neq 0$ only if $q = r$). We further simplify the Tucker decomposition by setting the factors dimensions $P, Q$ and $R$ to $\tilde{D}_k$, a single interpretable hyperparameter equal to the dimension of the *shared* key/query space that controls the amount of compression of the decomposition into collaborative heads. These changes lead to a special case of Tucker decomposition called the canonical decomposition, also known as CP or PARAFAC (Harshman, 1970) in the literature (Kolda & Bader, 2009). Fix any positive rank $R$. The decomposition yields:

$$\mathbf{T} \approx \sum_{r=1}^{R} \boldsymbol{a}_r \circ \boldsymbol{b}_r \circ \boldsymbol{c}_r =: [\![\boldsymbol{A}, \boldsymbol{B}, \boldsymbol{C}]\!], \qquad (11)$$

with $\boldsymbol{A} \in \mathbb{R}^{I \times R}$, $\boldsymbol{B} \in \mathbb{R}^{J \times R}$ and $\boldsymbol{C} \in \mathbb{R}^{K \times R}$.

What is remarkable is that the above can be used to express any (trained) attention layer parametrized by $\{\boldsymbol{W}_Q^{(i)}, \boldsymbol{b}_Q^{(i)}, \boldsymbol{W}_K^{(i)}, \boldsymbol{b}_K^{(i)}\}_{i \in [N_h]}$ as a collaborative layer. In particular, if we apply the decomposition to the stacked heads $\mathbf{W}_{QK}$ we obtain the three matrices $[\![\boldsymbol{M}, \tilde{\boldsymbol{W}}_Q, \tilde{\boldsymbol{W}}_K]\!]$ that define a collaborative attention layer: the mixing matrix $\boldsymbol{M} \in \mathbb{R}^{N_h \times \tilde{D}_k}$, as well as the key and query projection matrices $\tilde{\boldsymbol{W}}_Q, \tilde{\boldsymbol{W}}_K \in \mathbb{R}^{D_{in} \times \tilde{D}_k}$.

---

[3] $\circ$ represents the vector outer product

On the other hand, biases can be easily dealt with based on the content/context decomposition of eq. (3), by storing for each head the vector

$$\boldsymbol{v}_i = \boldsymbol{W}_K^{(i)} \boldsymbol{b}_Q^{(i)} \in \mathbb{R}^{D_{in}}. \qquad (12)$$

With this in place, the computation of the (unscaled) attention score for the $i$-th head is given by:

$$\left( \boldsymbol{X}\boldsymbol{W}_Q^{(i)} + \mathbf{1}_{T \times 1} \boldsymbol{b}_Q^\top \right) \left( \boldsymbol{Y}\boldsymbol{W}_K^{(i)} + \mathbf{1}_{T \times 1} \boldsymbol{b}_K^\top \right)^\top \approx \boldsymbol{X}\tilde{\boldsymbol{W}}_Q \operatorname{diag}(\boldsymbol{m}_i) \tilde{\boldsymbol{W}}_K^\top \boldsymbol{Y}^\top + \mathbf{1}_{T \times 1} \boldsymbol{v}_i^\top \boldsymbol{Y}^\top, \quad (13)$$

where $\boldsymbol{m}_i$ is the $i$-th row of $\boldsymbol{M}$. If $\tilde{D}_k \geq D_k$ the decomposition is exact (eq. (11) is an equality) and our collaborative heads layer can express any concatenation-based attention layer. We also note that the proposed re-parametrization can be applied to the attention layers of many transformer architectures, such as the ones proposed by Devlin et al. (2019); Sanh et al. (2019); Lan et al. (2020).

### 3.4 PARAMETER AND COMPUTATION EFFICIENCY

Collaborative MHA introduces weight sharing across the key/query projections and decreases the number of parameters and FLOPS. While the size of the heads in the standard attention layer is set to $d_k = 64$ and the key/query layers project into a space of dimension $D_k = N_h d_k$, the shared key/query dimension $\tilde{D}_k$ of collaborative MHA can be set freely. According to our experiments in Section 4 (summarized in Table 1), a good rule of thumb when transforming a *trained* MHA layer to collaborative is to set $\tilde{D}_k$ to half or one third of $D_k$. When training from scratch, $\tilde{D}_k$ can even be set to $1/4$-th of $D_k$.

Table 1: Comparison of a layer of concatenate vs. collaborative MHA with chosen $\tilde{D}_k$ to give negligible performance difference. T=128.

| | train FairSeq §4.1 | | re-param. HuggingFace §4.2 | |
|---|---|---|---|---|
| | concat. | collab. | concat. | collab. |
| $D_k \to \tilde{D}_k$ | $512 \to 128$ | | $768 \to 256$ | |
| Params ($\times 10^6$) | 1.05 | 0.66 | 2.36 | 1.58 |
| FLOPS ($\times 10^8$) | 1.51 | 1.09 | 3.27 | 2.65 |
| inference (ms) | 0.99 | 0.81 | 1.71 | 1.65 |

**Parameters.** Collaborative heads use $(2D_{in} + N_h)\tilde{D}_k$ parameters, as compared to $2D_{in}D_k$ in the standard case (ignoring biases). Hence, the compression ratio is $\approx D_k/\tilde{D}_k$, controlled by the shared key dimension $\tilde{D}_k$. The collaborative factorization introduces a new matrix $\boldsymbol{M}$ of dimension $N_h \times \tilde{D}_k$. Nevertheless, as the number of heads is small compared to the hidden dimension (in BERT-base $N_h = 12$ whereas $D_{in} = 768$), the extra parameter matrix yields a negligible increase as compared to the size of the query/key/values matrices of dimension $D_{in} \times D_k$.

**Computational cost.** Our layer decomposes two matrices into three, of modulable dimensions. To compute the attention scores between $T$ tokens for all the $N_h$ heads, collaborative MHA requires $2T(D_{in} + N_h)\tilde{D}_k + T^2 N_h \tilde{D}_k$ FLOPS, while the concatenation-based MHA uses $2TD_{in}D_k + T^2 D_k$ FLOPS. Assuming that $D_{in} \gg N_h = \mathcal{O}(1)$ (as is common in most implementations), we obtain a theoretical speedup of $\Theta(D_k/\tilde{D}_k)$. However in practice, having two matrix multiplications instead of a larger one makes our implementation slightly slower, if larger multiplications are supported by the hardware.

## 4 EXPERIMENTS

The goal of our experimental section is two-fold. First, we show that concatenation-based MHA is a drop-in replacement for collaborative MHA in transformer architectures. We obtain a significant reduction in the number of parameters and number of FLOPS without sacrificing performance on a Neural Machine Translation (NMT) task with an encoder-decoder transformer. Secondly, we verify that our tensor decomposition allows one to reparametrize *pre-trained* transformers, such as BERT (Devlin et al., 2019) and its variants. To this end, we show that collaborative MHA performs on par with its concatenation-based counter-part on the GLUE benchmark (Wang et al., 2018) for Natural Language Understanding (NLU) tasks, even without retraining.

The NMT experiments are based on the FairSeq (Ott et al., 2019) implementation of transformer-base by Vaswani et al. (2017). For the NLU experiments, we implemented the collaborative MHA layer as an extension of the Transformers library (Wolf et al., 2019). The flexibility of our layer allows it to be applied to most of the existing transformer architectures, either at pre-training or after fine-tuning

| | BLEU ↑ | | params (×10⁶) | | time (h) | |
|---|---|---|---|---|---|---|
| $D_k$ | concat. | collab. | concat. | collab. | concat. | collab. |
| 512 | 27.40 | 27.58 | 60.9 | 61.0 | 18.0 | 21.0 |
| 256 | 27.10 | 27.41 | 56.2 | 56.2 | 17.3 | 19.0 |
| 128 | 26.89 | 27.40 | 53.8 | 53.8 | 17.3 | 18.4 |
| 64 | 26.77 | 27.31 | 52.6 | 52.7 | 16.9 | 17.9 |

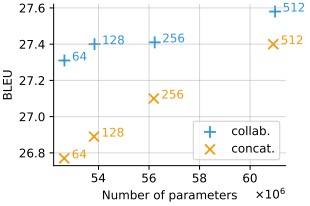
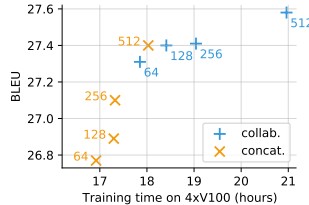

Figure 3: Comparison of the BLEU score on WMT14 EN-DE translation task for an encoder-decoder transformer-base (Vaswani et al., 2017) using collaborate vs. concatenate heads with key/query dimension $D_k$. We visualize performance as a function of number of parameters (*middle*) and training time (*right*). Collaborative attention consistently improves BLEU score, $D_k$ can be decreased by a factor of 4 without drop in performance.

using tensor decomposition. We use the tensor decomposition library Tensorly (Kossaifi et al., 2019) with the PyTorch backend (Paszke et al., 2017) to reparameterize pre-trained attention layers. Our code and datasets are publicly available[4] and all hyperparameters are specified in the Appendix.

## 4.1 COLLABORATIVE MHA FOR NEURAL MACHINE TRANSLATION

We replace the concatenation-based MHA layers of an encoder-decoder transformer by our collaborative MHA and evaluate it on the WMT14 English-to-German translation task. Following (Vaswani et al., 2017), we train on the WMT16 train corpus, apply checkpoint averaging and report compound split tokenized BLEU. We use the same hyperparameters as the baseline for all our runs. Results are shown in Figure 3. Our run of the original base transformer with $N_h = 8$ heads and $D_k = 512$ key/query total dimensions achieves 27.40 BLUE (instead of 27.30).

As observed in the original paper by Vaswani et al. (2017), decreasing the key/query head size $d_k$ degrades the performance (× in Figure 3). However, with collaborative heads (+ in Figure 3), the shared key/query dimension can be reduced by 4× without decreasing the BLEU score. As feed-forward layers and embeddings are left untouched, this translates to a 10% decrease in number of parameters for a slight increase in training time. When setting a total key/query dimension of $D_k = 64$, corresponding to $d_k = 8$ dimensions per head, the classic MHA model suffers a drop of 0.6 BLEU points, meanwhile the collaborative MHA stays within 0.1 point of the transformer-base model using concatenation.

We conclude that sharing key/query projections across heads allows attention features to be learned and stored only once. This weight sharing enables decreasing $D_k$ without sacrificing expressiveness.

## 4.2 RE-PARAMETRIZE A PRE-TRAINED MHA INTO COLLABORATIVE MHA

We turn to experiments on Natural Language Understanding (NLU) tasks, where transformers have been decisive in improving the state-of-the-art. As pre-training on large text corpora remains an expensive task, we leverage the post-hoc reparametrization introduced in Section 3.3 to cast already pre-trained models into their collaborative form. We proceed in 3 steps for each GLUE task (Wang et al., 2018). First, we take a pre-trained transformer and fine-tune it on each task individually. Secondly, we replace all the attention layers by our collaborative MHA using tensor decomposition to compute $\tilde{W}_Q$, $\tilde{W}_K$ and $M$ and re-parametrize the biases into $v$. This step only takes a few minutes as shown in Figure 4. Finally, we fine-tune the compressed model again and evaluate its performance.

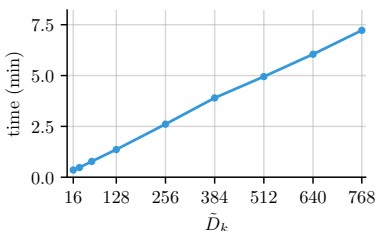

Figure 4: Time to decompose BERT-base from $D_k = 768$ to $\tilde{D}_k$.

We experiment with a pre-trained BERT-base model (Devlin et al., 2019). We also repurpose two variants of BERT designed to be more parameter efficient: ALBERT (Lan et al., 2020), an improved

---

[4]https://github.com/...

Table 2: Performance of collaborative MHA on the GLUE benchmark (Wang et al., 2018). We report the median of 3 runs for BERT (Devlin et al., 2019), DistilBERT (Sanh et al., 2019) and ALBERT (Lan et al., 2020) with collaborative heads and different compression controlled by $\tilde{D}_k$. Comparing the original models ($D_k = 768$) with their compressed counter part shows that the number of parameters can be decreased with less than 1.5% performance drop (*gray rows*).

| Model | $\tilde{D}_k$ | params | CoLA | SST-2 | MRPC | STS-B | QQP | MNLI | QNLI | RTE | **Avg.** |
|---|---|---|---|---|---|---|---|---|---|---|---|
| **BERT-base** | - | 108.3M | 54.7 | 91.7 | 88.8/83.8 | 88.8/88.7 | 87.6/90.8 | 84.1 | 90.9 | 63.2 | 83.0 |
| | 768 | 108.5M | 56.8 | 90.1 | 89.6/85.1 | 89.2/88.9 | 86.8/90.2 | 83.4 | 90.2 | 65.3 | 83.2 |
| | 384 | 101.4M | 56.3 | 90.7 | 87.7/82.4 | 88.3/88.0 | 86.3/90.0 | 83.0 | 90.1 | 65.3 | 82.5 |
| | 256 | 99.0M | 52.6 | 90.1 | 88.1/82.6 | 87.5/87.2 | 85.9/89.6 | 82.7 | 89.5 | 62.5 | 81.7 |
| | 128 | 96.6M | 43.5 | 89.5 | 83.4/75.2 | 84.5/84.3 | 81.1/85.8 | 79.4 | 86.7 | 60.7 | 77.6 |
| **DistilBERT** | - | 66.4M | 46.6 | 89.8 | 87.0/82.1 | 84.0/83.7 | 86.2/89.8 | 81.9 | 88.1 | 60.3 | 80.0 |
| | 384 | 62.9M | 45.6 | 89.2 | 86.6/80.9 | 81.7/81.9 | 86.1/89.6 | 81.1 | 87.0 | 60.7 | 79.1 |
| **ALBERT** | - | 11.7M | 58.3 | 90.7 | 90.8/87.5 | 91.2/90.8 | 87.5/90.7 | 85.2 | 91.7 | 73.7 | 85.3 |
| | 512 | 11.3M | 51.1 | 86.0 | 91.4/88.0 | 88.6/88.2 | 87.2/90.4 | 84.2 | 90.2 | 69.0 | 83.1 |
| | 384 | 11.1M | 40.7 | 89.6 | 82.3/71.1 | 86.0/85.6 | 87.2/90.5 | 84.4 | 90.0 | 49.5 | 77.9 |

transformer with a single layer unrolled, and DistilBERT (Sanh et al., 2019) a smaller version of BERT trained with distillation. We report in Table 2 the median performance of 3 independent runs of the models on the GLUE benchmark (Wang et al., 2018).

We first verify that tensor decomposition without compression ($\tilde{D}_k = D_k = 768$) does not alter performance. As shown in Table 2, both BERT-base and its decomposition performs similarly with an average score of 83.0% and 83.2% respectively. We should clarify that, for consistency, we opted to re-finetune the model in all cases (even when $\tilde{D}_k = D_k$), and that the slight score variation disappears without re-finetuning. Nevertheless, even with re-finetuning, reparametrizing the attention layers into collaborative form is beneficial in 4 out of the 8 tasks, as well as in terms of the average score.

We then experiment with compressed decomposition using a smaller $\tilde{D}_k$. Comparing the original models with their well-performing compressed counterpart (gray rows) shows that the key/query dimension of BERT and DistilBERT can be reduced by $2\times$ and $3\times$ respectively without sacrificing more than 1.5% of performance. This is especially remarkable given that DistilBERT was designed to be a parameter-efficient version of BERT. It seems that ALBERT suffers more from compression, but the dimension can be reduced by a factor $1.5\times$ with minor performance degradation. We suspect that unrolling the same attention layer over the depth of the transformer forces the heads to use different projections and decreases their overlap, decreasing the opportunity for weight-sharing. Our hypothesis is that better performance may be obtained by pre-training the whole BERT architecture variants from scratch.

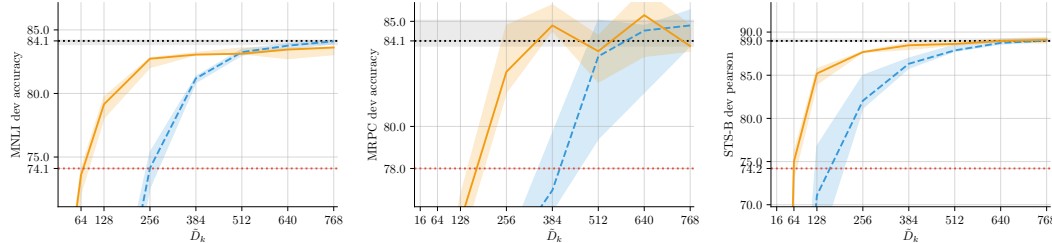

Figure 5: Performance on MNLI, MRPC and STS-B datasets of a ···· fine-tuned BERT-base model, ─ decomposed with collaborative heads of compressed dimension $\tilde{D}_k$ (*horizontal axis*). ─ Repeating fine-tuning after compression can make the model recover the original performance when compression was drastic. The ···· GLUE baseline gives a reference for catastrophic failure.

**Recovering from compression with fine-tuning.** We further investigate the necessity of the second fine-tuning—step 3 of our experimental protocol—after the model compression. Figure 5 shows the performance of BERT-base on 3 GLUE tasks for different compression parameters $\tilde{D}_k$ with

and without the second fine-tuning. We find that for compression up to $1.5\times$ (from $D_k = 768$ to $\tilde{D}_k = 512$), the re-parametrization is accurate and performance is maintained without fine-tuning again. Further compressing the model starts to affect performance. Nevertheless, for compression by up to $3\times$ (to $\tilde{D}_k = 256$), this loss can readily be recovered by a second fine-tuning (*in orange*).

## 5 CONCLUSION

This work showed that trained concatenated heads in multi-head attention models can extract redundant query/key representations. To mitigate this issue, we propose to replace concatenation-based MHA by collaborative MHA. When our layer is used as a replacement for standard MHA in encoder/decoder transformers for Neural Machine Translation, it enables the decrease of effective individual head size from $d_k = 64$ to 8 without impacting performance. Further, without pre-training from scratch, switching a MHA layer to collaborative halves the number of FLOPS and parameters needed to compute the attentions score affecting the GLUE score by less than 1.5%.

Our model can impact every transformer architecture and our code (publicly available) provides post-hoc compression of already trained networks. We believe that using collaborative MHA in models pre-trained from scratch could force heads to extract meaningful shared query/key features. We are curious if this would translate to faster pre-training, better performance on downstream tasks and improved interpretability of the attention mechanism.

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

# Supplementary Material

## A  HYPERPARAMETERS FOR NEURAL MACHINE TRANSLATION EXPERIMENTS

Our implementation is based on Fairseq implementation Ott et al. (2019). We report in the following tables the specification of the architecture. We used the default hyperparameters if they are not specified below.

| Transformer architecture parameters | |
|---|---|
| dataset | wmt16_en_de_bpe32k |
| architecture | transformer_wmt_en_de |
| layers | 6 |
| heads | 8 |
| hidden-dim | 512 |
| collaborative-heads | "encoder_cross_decoder" or "none" |
| key-dim | 64, 128, 256, 512 |
| share-all-embeddings | True |
| optimizer | adam |
| adam-betas | (0.9, 0.98) |
| clip-norm | 0.0 |
| lr | 0.0007 |
| min-lr | 1e-09 |
| lr-scheduler | inverse_sqrt |
| warmup-updates | 4000 |
| warmup-init-lr | 1e-07 |
| dropout | 0.1 |
| weight-decay | 0.0 |
| criterion | label_smoothed_cross_entropy |
| label-smoothing | 0.1 |
| max-tokens | 3584 |
| update-freq | 2 |
| fp16 | True |

Table 3: Hyperparameters for the NMT experiment.

## B  HYPERPARAMETERS FOR NATURAL LANGUAGE UNDERSTANDING EXPERIMENTS

We use standard models downloadable from HuggingFace repository along with their configuration.

| Models | | |
|---|---|---|
| BERT-base | Devlin et al. (2019) | bert-base-cased |
| DistilBERT | Sanh et al. (2019) | distilbert-base-cased |
| ALBERT | Lan et al. (2020) | albert-base-v2 |

We use HuggingFace default hyperparameters for GLUE fine-tuning in all our runs. We train with a learning rate of $2 \cdot 10^{-5}$ for 3 epochs for all datasets except SST-2 and RTE where we train for 10 epochs. In preliminary experiments, we tried to tune the tensor decomposition tolerance hyperparameter among $\{10^{-6}, 10^{-7}, 10^{-8}\}$ but did not see significant improvement and kept the default $10^{-6}$ for all our experiments.

| GLUE fine-tuning hyperparameters | |
| --- | --- |
| Number of epochs | 3 for all tasks but 10 for SST-2 and RTE |
| Batch size | 32 |
| Learning rate | 2e-5 |
| Adam $\epsilon$ | 1e-8 |
| Max gradient norm | 1 |
| Weight decay | 0 |
| Decomposition tolerance | 1e-6 |

