# OpenReview forum: "Multi-Head Attention: Collaborate Instead of Concatenate"
_ICLR.cc/2021/Conference — Reject_

### Official Review · AnonReviewer1 · 2020-10-27
**A nice paper, but has problems with supporting the claims and some parts are misleading**

**Rating:** 6
**Confidence:** 5

**Review:**

The paper investigates the over-parameterization of attention heads in Transformer’s multi-head attention. The authors show that query-key projections are redundant because trained concatenated heads tend to compute their attention patterns on common features. They propose a reparameterization of multi-head attention allowing the parameters of queries and keys to be shared between heads: this is called “collaborative attention”. This attention can be applied either instead of the standard attention during training, or as a drop-in replacement for an already trained model. To use as a drop-in replacement, the method requires to use tensor decomposition and subsequent model fine-tuning.

------------------------------------------------------------------------------------------------------
Strengths

1) a nice analysis of PCA components showing that individual heads are not low-rank, but their concatenation is.
2) the paper is overall clear and the method is explained well.

------------------------------------------------------------------------------------------------------
Weaknesses

(main) While the main contribution is a more efficient attention layer without a significant drop in performance, this claim is not supported empirically. Since the method operated only within attention layers (reduces only dimensions of queries and keys), in terms of efficiency/quality trade-off it should be compared to other methods, e.g. simple head pruning. While the paper does not provide such comparison, it is clear from the results that the simple head pruning is likely to be superior (and is simpler implementation-wise).

Namely,
1) when used as a drop-in replacement, the method is more complicated than a simple head pruning, but not more effective. E.g., the proposed method reduces query/key dimension by a factor of 1.5/2/3 and keeps everything else intact. With simple head pruning, about 50% of the heads can be removed without sacrificing quality: in addition to queries and keys, this also halves values and the output projection matrix.
2) when used in training (see the MT experiments), the results are also not better than post-hoc head pruning. The argument to support this new approach could be that the model is smaller in training, but since the parameter reduction is only in queries and keys, the decrease in the number of parameters is negligible for the whole model. For example, keeping the same quality it reduces the number of parameters from 6.1*10^6 to 5.4*10^6, which is not going to make a large difference.

(minor) As a side contribution, the authors claim to report the discrepancy between the theory and implementation of attention layers. Namely, while the original definition of attention layers does not have biases in linear projections for q/k/v in attention, the authors claim that implementations contain the bias terms, and spend some time showing how to model the biases in key and query layers properly.

However, the original Transformer implementation (tensor2tensor) does *not* have biases by default. This means that the authors are probably referring to some specific implementation, different from the original one provided by the Transformer’s authors. Therefore, I can not consider this as a contribution and think that this part is misleading for a reader.

P.S. Here is the tensor2tensor code I was referring to: https://github.com/tensorflow/tensor2tensor/blob/5f9dd2db6d7797162e53adf152310ed13e9fc711/tensor2tensor/layers/common_attention.py#L4416

------------------------------------------------------------------------------------------------------
Overall recommendation

Overall, I can not recommend accepting this paper. While there are some parts of the paper that I like, the main claim is not supported empirically: both in terms of baselines and the overall decrease in the number of parameters. Additionally, the part with the biases in attention implementation is misleading.

------------------------------------------------------------------------------------------------------
Update after author response
------------------------------------------------------------------------------------------------------

1) Context and content attention

Thank you for updating and saying that only some of the implementations include bias! I think now this part is not misleading and can be of interest. I still have some concerns that this part does not fit the whole story very well - but this is the matter of taste. In the current state, I think it is ok :)

2) On the comparison with head pruning and on the paper going beyond practical realm.

I agree with your comments, but I do think you should make it very clear in the paper. In the current state, the paper tried to make practical contributions and, since they mostly do not hold (e.g., head pruning is simpler in practice), it's hard to appreciate the paper's value. I think you need to modify the things you highlight, and with proper discussion it would be much better. For example, if you state explicitly that in practice pruning may be simpler, but your results say/illustrate something other than practical applications. You won't lose because of it; in fact, I think the opposite.

Overall, I think the paper has improved during the discussion period. In a hope that the authors address my later comments and discuss the pruning in the text, I'm raising the score.

---

> ### Author Response · Authors · 2020-11-20
> **Authors' answer to Reviewer 1**
>
> We would like to thank you for your thorough review and time you invested into reading our work. We address both of your concerns below.
>
> **On the comparison with head pruning.**
> We agree that head pruning is a more effective compression method for practitioners.
> However, we see pruning as an indicator that there is an overhead in the current MHSA. Pruning is not the solution (it is fixing the problem post-facto rather than from scratch). On the other hand, we provide evidence that the overhead within the MH mechanism can be reduced from scratch (rather than as a post-processing). We hope that our findings is a first step towards an efficient MH mechanism that does not result in degenerated heads.
>
>
> Whereas at this point the efficiency benefits are not impressive, our approach:
> * presents a principled/elegant way to reduce inefficiencies rather than relying on post-hoc processing.
> * we hope that our ideas (seeing MHSA as tensor decomposition) will lead to eventual further improvements -- especially when coupled with more efficient GPU implementations.
>
>
> Thank you for pointing out that the original code of the transformer does not use biases in the linear projections. It seems that enabling the biases by default is a common pitfall of other implementations of Transformers. For instance [BERT](https://github.com/google-research/bert/blob/master/modeling.py#L674), [fairseq](https://github.com/pytorch/fairseq/blob/a48f235636557b8d3bc4922a6fa90f3a0fa57955/fairseq/model_parallel/modules/multihead_attention.py#L30) and [HuggingFace](https://github.com/huggingface/transformers/blob/master/src/transformers/models/bert/modeling_bert.py#L221) implementations enable biases by default. We updated Section 2.2 to reflect this nuance but the content/context decomposition remains an important contribution for BERT reparametrization.

---

### Official Review · AnonReviewer2 · 2020-10-28
**Interesting collaborative MHA mechanism**

**Rating:** 5
**Confidence:** 3

**Review:**

This paper presents an interesting collaborative MHA to enable heads to share projections, which can be easily applied to most existing transformer-based models, including NMT and pre-training models. With using the new collaborative MHA, the number of parameters and FLOPs can be decreased.

The PCA-based analysis of the query/key matrices is interesting and impressive, which motivates the newly proposed MHA. The authors also propose a Tensor Decomposition based method to easily convert MHA to its collaborative version without retraining.

The paper is well-written and organized, the experiments are thorough. However, I have several concerns:
1)	In the Table 1, it will be more convincing, if the run time on CPU and GPU can be provided.
2)	All the pre-trained models (BERT/DistilBERT/ALBERT) are evaluated on the GLUE dataset, the experiments on more challenging tasks like QA (e.g. SQuAD 1.1/2.0) should be added.
3)	The proposed method seems to be not effective for pre-trained models, e.g. when the number of parameters is decreased from 108.5M to 96.6M, this reduction of model size is not that big, while the average score decreases from 83.2 to 77.6.
4)	It would be interesting to add more analysis on the patterns of learned mixing matrix M for different tasks.
5)	In the Figure 3, the training time of the standard MHA with D_k = 256 is 0.0?

---

> ### Author Response · Authors · 2020-11-20
> **Authors' answer to Reviewer 2**
>
> We would like to thank you for your time and for your interesting remarks on our work.
>
> 1. We followed your suggestion and added the inference time on a V100 of our layer in Table 1. We see an improvement of 18% and 3.5% in both of our code bases.
> 2. We agree that additional numerical results could be useful. Our current set of experiments were selected to support our theoretical insights, rather than establishing the general efficiency of collaborative heads across a wide-range of scenarios. We preferred including an ablation rather than an additional task.
> 3. We included $\tilde D_k = 128$ to illustrate the limitations of compression. Still, selecting $\tilde D_k = 256$ (Table 2, grey-line) performs within 1.5% of the original model and yields non-negligible parameters reductions of 8.5%. The fact that the performance drop occurs between 256 and 128 can also be predicted by the decomposition of the key/query matrices (Figure 1 right), as 256 principal components are needed to capture the variance (the red curve falls below 1 for fewer components).
> 4. We tried to study the patterns learned by the mixing matrix M but we have not yet found any interesting behavior. We still think that this is an interesting direction, as well as trying the full Tucker decomposition.
> 5. Thank you for pointing out the 0.0 running timing error. There was a small mistake in the table generation, but the plots were correct.
>
> We hope that we addressed your concerns and welcome any new suggestion.

---

> > ### Comment · AnonReviewer2 · 2020-11-24
> > **some of my concerns are addressed**
> >
> > Dear authors,
> >
> > Thanks for your explanation. Your explanation addressed some of my concerns, while, regarding the concern 2&3, if the ultimate goal of this work is to achieve an effective BERT compression method, it is important to verify it on various application scenarios, and the final results are not that competitive in comparison to the other BERT compression methods, e.g. KD-based, Pruning-based, or Quantization-based methods. The DistilBERT model is already not a strong baseline, please refer to the recent advances, e.g. MobileBERT, TinyBERT, MiniLM, etc.

---

> > > ### Author Response · Authors · 2020-11-24
> > > **Our Multi-Head Attention can be trained from scratch rather than distilled or compressed post hoc**
> > >
> > > Dear reviewer,
> > >
> > > Thank you for your answer. The main purpose of our method is not to compress a pretained model, we see this significant part as a validation of our theory showing that heads projections are shared. Our main contribution is to propose a more parameter efficient Multi-Head mechanism that can be trained from scratch. On the other hand, pruning is only a post hoc fix for the degenerated heads and the interesting works that you mention indeed display better compression rate but are trained by distillation from a teacher network.

---

> > > > ### Comment · AnonReviewer2 · 2020-11-25
> > > > **Please Add References on Tensor Decomposition-based Deep Model Compression**
> > > >
> > > > Applying tensor decomposition to neural networks for efficient parameter sharing is not a new topic, and an important reference is not cited, please check it
> > > >
> > > > COMPRESSION OF DEEP CONVOLUTIONAL NEURAL NETWORKS FOR FAST AND LOW POWER MOBILE APPLICATIONS, ICLR-2016

---

> > > > > ### Author Response · Authors · 2020-11-25
> > > > > **We added the reference**
> > > > >
> > > > > Dear Reviewer 2,
> > > > >
> > > > > Thank you for pointing this interesting work that uses the Tucker decomposition to compress already trained CNNs. We added the reference to this related work. We are not aware of Tucker decomposition applied to multi-head attention models and the insight that key/query projections are shared is novel. Inspired by the heads decomposition, we propose a new MHA layer that can be trained from scratch.

---

### Official Review · AnonReviewer3 · 2020-10-29
**Well motivated but weak results**

**Rating:** 5
**Confidence:** 4

**Review:**

This paper analyzes the multi-head attention in transformers and suggests to use collaboration instead of concatenation of multiple heads. Empirical results on WMT’16 English-German demonstrates that the proposed approach reduces the of parameters without sacrificing performance. Further experiments on pre-trained BERT models also demonstrate its efficiency.  Overall, the paper is well motivated and provides a deep analysis of redundancy of the multi-head attention.

Here are a few detailed comments:

-- In practice, it seems the proposed approach didn’t speed up the training, even though it gives an improvement in terms of FLOPS. This may limit to the large scale of BERT pre-training.

-- From Figure 3 and Table 2, even reducing the $\hat{D_k}$ from 768 to 128, the total number of parameters of pretrained models (e.g. BERT-base) only reduces from 108.3M to 96.6M. However, the performance has dropped significantly (83.0 -> 77.6 GLUE score) in the BERT-base setting. If we take a closer look, in case of $\hat{D_k}$=768, the performance on the large tasks (e.g. MNLI) dropped significantly from 84.1 to 83.4 in terms of accuracy, and its improvement is from small tasks (e.g. RTE). It is better to have a deeper analysis and explanation.

===== Update after author response =====

If you look a deeper look into Transformers, in case of BERT, the attention block only takes around 25% of total parameters. It is that suspicious that collaborative MHA takes 18% less time in practical since it requires many factors e.g., GPU kernel fusion.

Regarding the performance on MNLI, from Fig 5, it shows that when D_k is larger than 512, MHA reaches to the baseline in terms of accuracy. Additional information from Tab 2, these models have more than 101.4M (in case of D_k = 384) which is almost the same as the original baseline. However, the performance on GLUE is dropped largely, thus it is hard to support papers' claims.

---

> ### Author Response · Authors · 2020-11-20
> **Authors' answer to Reviewer 3**
>
> We would like to thank the reviewer for their time and the insightful remarks they provided on our work.
>
> It is true that our method does not improve the training time. On the NMT experiment, both concat 512 and collab 128 reach the same BLEU score (27.40) in approximately the same training time (+2%), but with the latter having 15% fewer parameters. We do not see a reason for not applying this to BERT pre-training. Concerning speed at inference, Table 1 now includes timing measurements for the collab attention layer (following Reviewer 5’s advice). With our method, the multi-head self-attention mechanism takes 18% less time when trained from scratch ($D_k = 512 \to 128$ in FairSeq) and 3.5% less time when re-parameterized ($D_k = 768 \to 256$ in HuggingFace). Regarding parameter reduction: First off, it is important to stress that, whereas we focus only on the inefficiencies of MHA, the evaluated model contains many additional components (that we do not attempt to compress). Thus, the maximum possible parameter reduction is capped.
>
> Concerning BERT re-parametrization, we find that decreasing $D_k$ from 768 to 256 yields non-negligible parameters reduction of 8.5%. As the reviewer pointed out, too much compression can hurt and we included $\tilde D_k = 128$ to illustrate this limitation. Still, selecting $\tilde D_k = 256$ (grey-line) performs within 1.5% of the original model. The fact that the performance drop occurs between 256 and 128 can also be predicted by the decomposition of the key/query matrices (Figure 1 right), as 256 principal components are needed to capture the variance (the red curve falls below 1 for fewer components).
>
> We also clarify that for $\tilde D_k = 768$ (no compression), the performance remains identical on MNLI at 84.1 (see Figure 5, Left, blue curve), and re-finetuning the network changes the performance to 83.7. The issue could have been avoided by skipping the finetuning step. However, for consistency, we opted to retain the finetuning hyperparameters identical in all cases.
>
> Finally, as shown in Table 2, even though re-parametrization together with finetuning will slightly change the performance, the change is beneficial in 4 out of the 8 tasks and results in a small net gain overall.
>
> We added a more detailed analysis of the GLUE results in the revision of the manuscript.

---

### Official Review · AnonReviewer5 · 2020-11-08

**Rating:** 5
**Confidence:** 3

**Review:**

========================

Paper Summary:

This paper proposes a new form of multi-head attention. It can reduce parameters and FLOPs of Transformer models without performance loss on En-De translation. Moreover, for pre-trained language models, no large-scale pretraining is required to convert the attention. A tensor decomposition based method is proposed for the conversion.

==========================

Overall review

This paper challenges the widely adopted multi-head attention, asking the question whether the concatenation of multiple heads is the best way to fuse heads. The proposed method is well-motivated (Fig. 1), but the empirical performance does not show a clear advantage over the conventional way. More rigorous experiment is needed to justify this new model.

Pros

- the method is novel and well-motivated
- works for both original transformer and pre-trained language models

Cons

- improvements (in terms of evaluation metrics) is marginal to original MHA
- more translation / generation tasks should be evaluated
- does not measure empirical speed up at inference time

==========================

Questions / Suggestions

- In the abstract, the authors argued for over-parameterization. In fact, over-parameterized Transformers such as GPT-3 in fact achieves very strong performance. This is still an open question so might not be appropriate to say transformers suffers over-parameterization.
- In the end of introduction, Synthesizer model is mentioned but this part is not clearly explained in the rest of the paper.
- Figure 2 is somewhat confusing. I couldn't understand the relationship between (c) & (d) to the rest of the figure.

===========================

Minor Issues

- Figure 4 x-axis: 767 -> 768

---

> ### Author Response · Authors · 2020-11-20
> **Authors' answer to Reviewer 5**
>
> First off, we would like to thank you for your constructive comments.
>
> Thank you for bringing up that overparameterization might be beneficial for very large transformers. We updated the sentence in the new revision of the manuscript.
>
> We think our experiments are rigorous and we do not overclaim on the performance of our method. We agree that the presented re-parametrization results do not demonstrate a significant advantage compared to other compression methods. However, they illustrate well our theoretical insights by showing that current head concatenation schemes can be re-expressed in equivalent but more efficient forms. Furthermore, the NMT experiment demonstrates that one can reduce by a factor of 4 the key/query dimensions without decreasing the BLEU score, which is not marginal. We argue that achieving a constant factor improvement in a mathematically principled manner (i.e., by generalizing MHSA, without relying on post-hoc heuristics such as pruning) is a valid contribution, especially given the widespread use of multi-head self-attention layers.
>
> We also agree with the reviewer that additional numerical results could be useful. Our current set of experiments were selected with the mindset of supporting our theoretical insights, rather than establishing the general efficiency of collaborative heads across a wide-range of scenarios. We aim to consider additional tasks in a follow up work.
>
> We added the empirical speedup at inference time that you requested in Table 1 for the MHA layer with concatenation vs. collaboration. As seen, our layer is slightly faster at inference when configured with a $D_k$ that matches the original performance.
>
> We also included additional explanations to the relationship between content vs. context attention and Synthesizer. We have moved this discussion to Section 2.2.
>
> Figure 2: (c) and (d) are alternative mixing matrices where all entries are learned, as opposed to fixed to 0 and 1 for each separated head. We updated the caption to clarify this.
>
> Thank you for pointing out the typo 767 $\to$ 768.
>
> We hope that we addressed all your concerns and we would be happy to answer any other questions you might have.

---

### Author Response · Authors · 2020-11-20
**Thank you for your reviews**

We would like to thank the reviewers for the time they spend reading our work and for the interesting suggestions they made to improve it.

We summarize our answers below:

1. We reported the speed up at inference of our collaborative head attention layer in Table 1 (R2, R5). Collaborative heads attention yields an improvement of 18% inference time when trained from scratch ($D_k = 512 \to 128$ in FairSeq) and 3.5% less time when re-parameterized ($D_k = 768 \to 256$ in HuggingFace).
2. Both R3 and R2 were concerned by the drop in performance when re-parametrizing BERT from 768 to 128 dimensions. We included this data point to show what happens if one compresses too much and to demonstrate that the required dimension size can be  predicted based on a PCA decomposition: as seen in Figure 1 (right), close to 256 dimensions are needed to capture all the variance.
3. We agree that head pruning is also a viable compression method (R1). However, we see our contribution as orthogonal to pruning, as pruning is not addressing the real problem of MHA, only fixing it post-facto. On the contrary, we provide evidence that the overhead within the MH mechanism could be reduced in a more principled manner, by allowing heads to learn shared subspaces. Our experiments show that, without any loss in performance, allowing heads to collaborate reduces (i) the key/query dimension by a factor of 3 and 4, and (ii) the number of parameters of each MHA layer by 33% and 37%, respectively, when the original layer is re-parameterized or when the model is re-trained from scratch.
4.We clarified why bias reparametrization is needed (R1) and how content/context decomposition could be linked to Synthesizer (R5).

We would like to emphasize that, overall, our work goes beyond the strictly practical realm (of finding a way to overcome the over-parameterization post-training) by delving into the causes of current MHA inefficiencies. Specifically, by viewing MHA as tensor decomposition we arrive at a better understanding of how heads interact. We find that different heads in standard implementations relearn to focus on similar subspaces, and that this can be mitigated by allowing them to jointly learn a shared subspace. Our analysis brings modest (but non-negligible) improvements in number of parameters and speed at inference of MHA without affecting accuracy. Further, we argue that our insights provide a fresh look into self-attention, one of the most popular layers of modern deep learning.

We provide individual responses below.  We appreciate the reviewer and the AC's continued diligence on our correspondences.

---

### Decision · Program_Chairs · 2021-01-07
**Final Decision**

**Decision:**

Reject

**Comment:**

This paper proposes an interesting collaborative multi-head attention (MHA) method to enable heads to share projections, which can reduce parameters and FLOPs of transformer-based models without hurting performance on En-De translation tasks. For pre-trained language models, a tensor decomposition method is used to easily covert the original MHA to its collaborative version without retraining.

This paper receives 3 weak reject and 1 weak accept recommendations. On one hand, all the reviewers agree that the paper is well motivated and the proposed idea is interesting. On the other hand, all the reviewers also commented that the current empirical results and comparisons are weak, which are not enough to support the paper's main claim. From the current results, it is difficult to draw a conclusion that collaborative MHA is better.

Specifically, (i) From Table 2, it can be seen that the proposed method is not effective for pre-trained models, i.e., even if the model size is not reduced much, the performance can be dropped significantly. (ii) More experiments, such as QA, more translation/generated tasks will make this paper more convincing. (iii) More rigorous experiments are needed to justify the practical value of the proposed method. If the authors try to emphasize that they go beyond practical realm, then probably a careful re-positioning of the paper is needed, which may not be a trivial task.

The rebuttal unfortunately did not fully address the reviewers' main concerns. Therefore, the AC regrets that the paper cannot be recommended for acceptance at this time. The authors are encouraged to consider the reviewers' comments when revising the paper for submission elsewhere.